# Ratiometric nanothermometer in vivo based on triplet sensitized upconversion

Ming Xu[1], Xianmei Zou[1], Qianqian Su[1], Wei Yuan[1], Cong Cao[1], Qiuhong Wang[1], Xingjun Zhu[1], Wei Feng [1] & Fuyou Li[1]

Temperature is an essential factor that counts for living systems where complicated vital activities are usually temperature dependent. In vivo temperature mapping based on non-contact optical approach will be beneficial for revealing the physiological phenomena behind with minimized influence to the organism. Herein, a highly thermal-sensitive upconversion system based on triplet–triplet annihilation (TTA) mechanism is pioneered to indicate body temperature variation sensitively over the physiological temperature range. The temperature-insensitive $NaYF_4$: Nd nanophosphors with NIR emission was incorporated into the temperature-responsive TTA-upconversion system to serve as an internal calibration unit. Consequently, a ratiometric thermometer capable of accurately monitoring the temperature changes in vivo was developed with high thermal sensitivity (~7.1% $K^{-1}$) and resolution (~0.1 K).

---

[1] Department of Chemistry & State Key Laboratory of Molecular Engineering of Polymers, Fudan University, Shanghai 200433, China. Correspondence and requests for materials should be addressed to W.F. (email: fengweifd@fudan.edu.cn) or to F.L.(email: fyli@fudan.edu.cn)

The luminescent nanothermometry has received much attention in recent years because it has a broad range of applications involving nanomedicine, microfluidics, nanoelectronics, and integrated photonic devices[1–3]. The development of such highly sensitive nanotermometer is very important in view of its great potential to revolutionize relevant areas, especially in the part of diagnosis and therapy[4]. Recently, temperature monitoring in vivo has been proposed as a useful tool for the studies of physiology, medical diagnosis, and controllable hyperthermia treatment[5–10]. The contactless thermometry based on luminescence imaging provides a noninvasive and observable approach to the mapping of body temperature[1–3]. Notably, upconversion based on the anti-Stokes process that can avoid auto-fluorescence of biological system, is a promising technique for the development of thermometer in vivo[11–13]. In this context, some thermometers were developed based on the lanthanide-doped upconversion nanophosphors (UCNPs). However, the UCNPs-based thermometers generally showed moderate thermal sensitivity (<1.6% K$^{-1}$) and resolution (>0.5 K), as well as poor luminescence efficiency[14–18]. Therefore, a highly sensitive thermometer capable of monitoring the slight temperature variation in vivo is still urgently needed.

As the most effective anti-Stokes process, upconversion based on triplet–triplet annihilation (TTA) is potentially thermosensitive[19–23]. TTA-upconversion involves multiple energy transfer in the annihilator and sensitizer dyad. Intuitively, TTA-upconversion requires diffusion of the component chromophores, which is sensitive to small temperature changes[24–30]. Nevertheless, the example of thermometer in vivo based on the TTA-upconversion technique has not been reported, which is hampered by significant challenges such as irregular temperature response or low thermal sensitivity in the physiological circumstance, and serious concentration dependence.

Herein, we designed an optimized TTA dyad to explore thermometry in vivo. At higher temperature, the improvement in diffusion rate and collision probability of the chromophores led to sharply enhanced TTA-upconversion[31–33]. Together with the suppression of nonradiative deactivation in our design, the upconversion luminescence (TTA-UCL) was positively temperature-dependent over the physiological temperature range (Supplementary Figure 1). In order to minimize influence from biological environment and to enable concentration-independent output of indicating signals, the TTA dyad was encapsulated with a thermal-insensitive internal standard (Supplementary Figure 2)[34]. The reference showed NIR emission in the second biological window, which was also applicable for bioimaging[35,36]. Consequently, a ratiometric thermometer was achieved as TTA-Nd-NPs (Fig. 1). The potential use of TTA-Nd-NPs for ratiometric thermometry in vivo was demonstrated by the accurate measurement of temperature distributions in tissue, and the detection of temperature changes in mice caused by inflammation. The work makes great sense for a broad research areas of upconversion, thermometry, nanomedicine, and life science.

## Results

**Upgrades to TTA system: deactivation suppression.** In terms of screening a possible TTA system for temperature indication, the photo-stable and effective BODIPY & PtTPBP dyads are promising candidates[37–40]. Apparently, phenyl ring at the top of conventional BODIPY annihilators (e.g., BD) could freely rotate to dissipate energy, thus leading to pronounced thermal deactivation (Fig. 2a)[41]. Diffusion enhancement at higher temperature is a positive factor for the TTA-upconversion process (Supplementary Figure 1), while thermal deactivation is a negative factor. The competitive effect between deactivation and diffusion factors

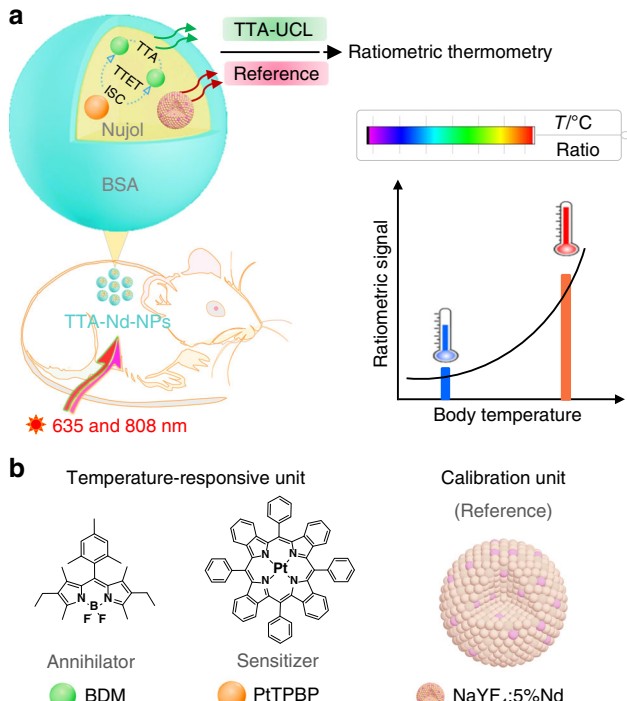

**Fig. 1** Schematic illustration of the TTA-Nd-NPs based ratiometric thermometry in vivo. **a** TTA-upconversion is sensitively responsive to temperature changes. With the assistance of an internal reference, the calibrated TTA-UCL signals become capable for accurate temperature monitoring in a small animal. **b** Chemical structures of the TTA chromophores containing BDM (TTA annihilator) and PtTPBP (TTA sensitizer), and schematic structure of the Nd$^{3+}$ nanophosphor (reference). BSA: bovine serum albumin, ISC: intersystem crossing, TTET: triplet–triplet energy transfer, TTA: triplet–triplet annihilation

would probably result in irregular thermal responses of the TTA system. As shown in Fig. 2b, a positive-to-negative transition at 40 °C was indicated with week TTA-UCL signal for BD & PtTPBP in nujol solvent. The TTA-UCL enhanced by positive diffusion effect was even overwhelmed by the negative quenching effect at higher temperature, which caused opposite thermal-response over 40 °C. Due to the quenching effect from thermal deactivation, the development of a sensitive TTA-thermometer in vivo is still challenging.

We took a simple but effective strategy to mitigate deactivation effect by using the rotation-suppressed BDM as the annihilator (Supplementary Figure 3). The structural modification had minor influence on the UV–Vis absorption, whereas it enabled a 1.6-fold improvement of the fluorescence (Supplementary Figure 4). Moreover, a remarkable mitigation in the thermal deactivation of fluorescence was observed (Fig. 2a), which was equivalent to the amplification of positive diffusion effect. Notably, our design pushed the limit of thermosensitive TTA system where phase transition or polymer-chain softening was generally required to enhance the diffusion effect[24,28,29,33]. Consequently, for the BDM & PtTPBP directly in liquid solvent, a continuous enhancement of TTA-UCL was achieved from 10 to 60 °C (Fig. 2b). Therefore, the discovery provided a new frontier in temperature-enhanced TTA system.

**Upgrades to TTA system: ratiometric calibration.** The thermal response of BDM & PtTPBP over physiological temperature range is highly desirable. Nevertheless, calibration of TTA-UCL is

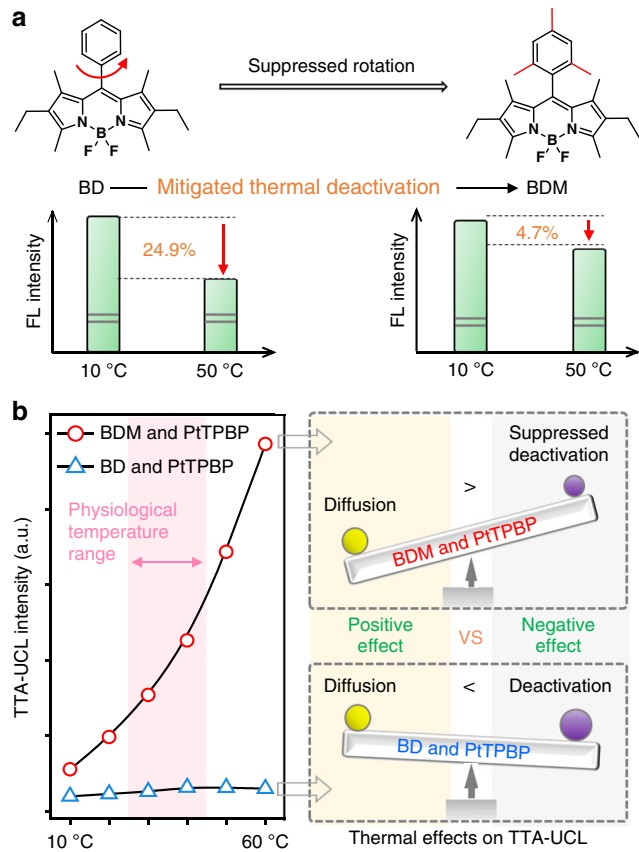

**Fig. 2** Strategy for achieving highly positive thermal response in a TTA system. **a** The structure modification and thus suppressed thermal deactivations of fluorescence. **b** The temperature responses of TTA-UCL signals emitted from BDM & PtTPBP and BD & PtTPBP, respectively. When negative deactivation effect is suppressed, the positive diffusion effect becomes more powerful to give a positive temperature-response trend with high sensitivity. A 635 nm laser (100 mW cm$^{-2}$) was used as the excitation source. The concentrations of BDM, BD, PtTPBP, and L-ascorbyl 6-palmitate in nujol solvent were $1\times10^{-3}$ M, $1\times10^{-3}$ M, $5\times10^{-5}$ M and 1 mg mL$^{-1}$, respectively

still necessary for thermometry in a complicated biosystem[42]. Therefore, a calibration unit was introduced into the TTA system in the present study. The TTA-Nd-NPs material was prepared through self-assembly of BDM & PtTPBP and β-NaYF$_4$: 5% Nd (Supplementary Figure 5) in a high pressure homogenizer. The calibration signal ($^4F_{3/2}$–$^4I_{11/2}$ of Nd$^{3+}$) had minimized overlap with the TTA-UCL, which could avoid the crosstalk effect (Supplementary Figure 2). The nanocomposite structure of organic–inorganic TTA-Nd-NPs was revealed in the transmission electron microscope images with a diameter of ~160 nm (Supplementary Figure 6). The hydrodynamic diameter of TTA-Nd-NPs was 165 nm, and the size kept unchanged within 10–50 °C (Supplementary Figure 7). In addition, no obvious toxicity of TTA-Nd-NPs material was revealed in the biocompatibility study (Supplementary Figure 8).

The TTA dyad BDM & PtTPBP in TTA-Nd-NPs gave rise to a green upconversion emission with lifetime of 170 μs (Supplementary Figure 9) and absolute quantum efficiency of 3.1% at room temperature (see Methods for details). A sustained downward trend in TTA-UCL intensity was observed as the TTA-Nd-NPs returned to room temperature through natural cooling, while the TTA-UCL intensities of control groups with constant temperatures remained unchanged all along. (Supplementary Figure 10). Moreover, the thermal sensitivity of TTA-

UCL was easily controllable by regulating the concentration ratio of sensitizer (Supplementary Figure 11). A quadratic-to-linear dependence between TTA-UCL intensity and 635 nm excitation power density was clearly observed, and the threshold excitation intensity was revealed to be 65 mW cm$^{-2}$ (Fig. 3a)[43]. Under low-power excitation below the threshold, triplets of the annihilators decay spontaneously to result in a quadratic dependence, which is a common phenomenon in the bimolecular TTA-upconversion system[43]. In addition, the luminescence intensity of Nd$^{3+}$ nanophosphors in TTA-Nd-NPs was linear dependent on the 808 nm excitation power density (Fig. 3a) with an absolute quantum

efficiency of 10.2% (see Methods for details). In the following experiments, the power densities of coupled 635 and 808 nm excitation light were therefore set beyond the threshold (>65 mW cm$^{-2}$) to ensure both of the luminescence processes in a linear regime. Herein, laser power density in the range of 100–200 mW cm$^{-2}$ was an optimal choice for excitation, which could enable power-independent signal output in the low-power region (Fig. 3a). With all above upgrades to TTA system, problems from irregular (or insensitive) temperature response and concentration dependence are solved in theory, and the concept of ratiometric probe based on TTA system is constructed.

**Ratiometric response of TTA-Nd-NPs.** Then, the thermo-sensitive properties of TTA-Nd-NPs were accurately investigated. The luminescence spectra of TTA-Nd-NPs were measured in a dual-channel test system as illustrated in Fig. 3b. In the measurement, the sample temperature was precisely controlled (±0.1 K) with a thermocouple placed in the solution of sample rather than around the surface of cuvette (Supplementary Figure 12). As the temperature increases, the TTA-UCL intensity enhanced rapidly to show a quadratic tendency (Fig. 3c). The calculated thermal sensitivity of TTA-UCL in the TTA-Nd-NPs (~12% K$^{-1}$) was far better than that of the reported thermosensitive NaYF$_4$: Yb, Er@NaLuF$_4$ UCNPs (~0.8% K$^{-1}$) within 10–50 °C (Supplementary Figure 13)[17]. In contrast, the emission from Nd$^{3+}$ nanocrystals in TTA-Nd-NPs slowly declined with a slop less than 0.03% K$^{-1}$ (Fig. 3c). Indeed, the Nd ion was not good for thermal response due to its low thermal sensitivity[44,45]. Attributed to the highly thermosensitive nature of our TTA system that was served as temperature-responsive unit, herein the thermal insensitive Nd$^{3+}$ nanocrystal could be designed to just serve as an internal standard. The absorption/emission of Nd$^{3+}$ nanocrystals showed no overlap with that of the TTA system, which enabled the unaffected performance of TTA system. Actually, calibration unit for the TTA system can also be upgraded in the future, for example employing other rare earth ions based core–shell nanoparticles that showed markedly thermal sensitivity, which is possibly beneficial for achieving a better performance in ratiometric thermometry[46,47].

In the biological system, the luminescence of exogenous probes generally depends on the specific bio-distributions (Supplementary Figure 14). Herein, the strategy of ratiometric nanocapsule was appropriately used to address this issue. For each individual TTA-Nd-NPs capsule, theoretically the luminescence ratio of TTA dyad and Nd$^{3+}$ nanocrystal maintains the same, ensuring the concentration-independent output[1-3]. The calibrated TTA-UCL signal output (ratio of $I_{\text{TTA-UCL}}$ / $I_{\text{Reference}}$) did not change with the concentration variations from the initial material (15 mg mL$^{-1}$) to a very dilute solution (0.9 mg mL$^{-1}$), even though the TTA-UCL signals were much decreased to show an obviously concentration-dependent signal output (Fig. 3d). The observations clearly demonstrated the importance of ratiometric probing

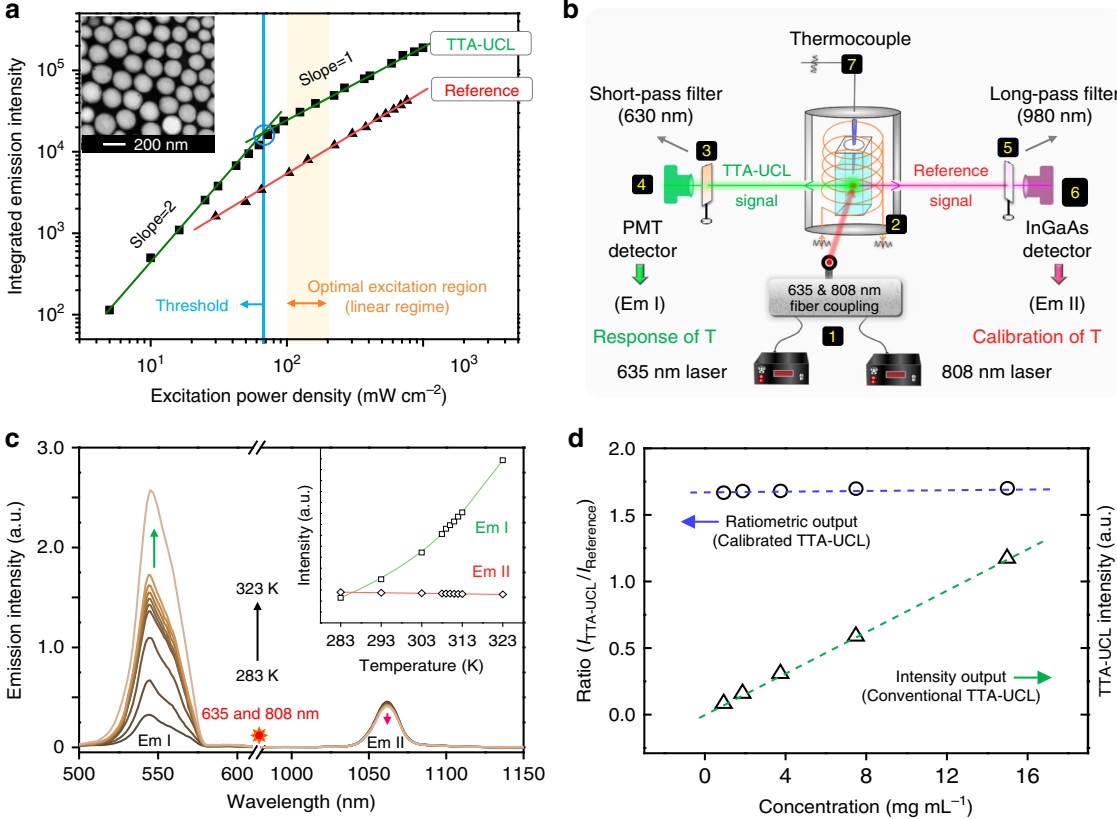

**Fig. 3** Properties investigation of TTA-Nd-NPs. **a** The dual-emissive properties of TTA-Nd-NPs at varying excitation power. TTA-UCL signal peaked at 540 nm and Reference signal peaked at 1060 nm were plotted in the emission profiles. Inset: the transmission electron microscope image of TTA-Nd-NPs after negative staining with sodium phosphotungstate. Scale bar is 200 nm. **b** Schematic illustration of the setup for calibrated TTA-UCL measurement under precisely controlled temperature. **c** The thermal sensitivity of TTA-Nd-NPs in aqueous solution. The spectra were measured with the excitation of coupled 635 nm (100 mW cm$^{-2}$) and 808 nm (100 mW cm$^{-2}$) laser. Inset: the integrated emission intensities of TTA-UCL (Em I, green line) and Reference signals (Em II, red line) as functions of temperature. **d** The concentration effects on the ratiometric probing method (blue line) and the intensity-based probing method (green line) at room temperature

method for TTA system. Furthermore, the pH factor that varied in different situations and tissues was also investigated, and the results under pH of 5.0–8.0 were almost uniform (Supplementary Figure 15). Moreover, the TTA-Nd-NPs probe was stable and with good reversibility (Supplementary Figure 16). These properties made the TTA-Nd-NPs suitable for temperature imaging in vivo without interference from the biological environment.

**Ratiometric thermometry in vivo based on TTA-Nd-NPs.** The standard curve of TTA-Nd-NPs for practical temperature evaluation in vivo was calibrated in the bioimaging system (Fig. 4a)[17,48]. The simulated condition in tissue phantom was an analogy of the subcutaneous injection in vivo (Supplementary Figure 17), and the data points was fitted as a following equation with $R^2 > 0.999$, Ratio ($I_{\text{TTA-UCL}}$ / $I_{\text{Reference}}$) = $0.00375T^2$ $-2.12032T+300.47099$ (Fig. 4b). The applicability of this calibration curve was then verified in the thermometry of a living mouse (Supplementary Figure 18). Actually, the TTA-Nd-NPs could reveal subcutaneous temperature in situ without physical contact, which was superior to the classical thermocouple that required skin puncture. Accordingly, the curve of thermal sensitivity [$S_R$ = (dRatio / d$T$) /Ratio] was shown in Supplementary Figure 19[1,47,49,50]. Based on these results, the thermal sensitivity was up to ~7.1% K$^{-1}$ with a high resolution of ~0.1 K, which represent the best results among all the thermometers developed for in vivo thermometry to date (Supplementary Table 1)[51].

Frankly, the weakness of this excellent TTA-Nd-NPs thermometer was revealed as the moderate penetration depth of green UCL signal (Supplementary Figure 20). Therefore, a thermosensitive TTA-UC system working in the NIR domain could be an attractive breakthrough for future applications in deep tissues. The NIR emissive sensitizer severing as reference could also simplify the systems in configuration[46,47].

Furthermore, the possible applications of our thermometer in vivo was explored. Generally, inflammation is interrelated with the physiological disorder in biological system which may cause temperature-related variations[4,52,53]. The inflammatory phenomena has been demonstrated in distinct processes, such as the ischemia symptoms, physical trauma, microbial infection, and drug induction[4,52,53]. Previously, the nanothermometry results revealed that the inflammatory process in ischemia experiments could lead to distinctive thermal dynamics in mice[4]. With these in mind, we intended to establish a chemically-induced arthritis with carrageenan and to monitor the inflammation-based temperature fluctuations directly (Supplementary Figure 21). Notably, we demonstrated that the inflammation in arthritis model was time-dependent and was accompanied with temperature variations (Supplementary Figure 22). To verify the protocol feasiblity, the possible influence of excitation laser on thermometry was also investigated[54]. Indeed, the low-level laser exposure in the present study could keep its influence on thermometry at a minimum (Supplementary Figure 23). These observations proved

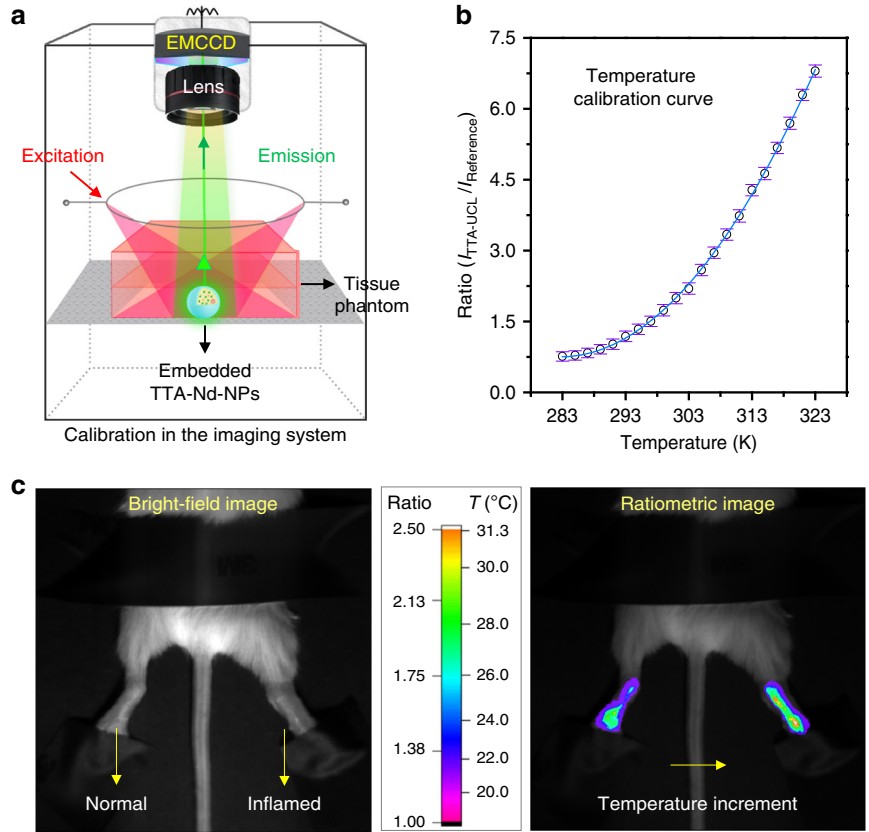

**Fig. 4** Ratiometric thermometry in vivo. **a** Schematic illustration of the setup for ratiometric thermometry in vivo. **b** The standard curve for temperature evaluation in vivo measured with TTA-Nd-NPs probe in tissue phantom. **c** Ratiometric thermometry based on TTA-Nd-NPs in an inflammatory mode. The bright-filed image showed arthritis in the swollen right leg of a Kunming mouse stimulated with carrageenan (1 wt%, 100 μL) while the left leg was normal as control. The ratiometric image revealed the luminescent ratios and evaluated temperature distributions in the two legs. The TTA-UCL and reference signals were acquired in channel I (485–575 nm) and channel II (980–1300 nm) of the in vivo imaging system, respectively

that our probe was promising for thermometry in the inflammation model.

Finally, the TTA-Nd-NPs based thermometer was tested for thermometry in the inflammatory mode. Herein, mild arthritis in the right leg of a mouse was induced with carrageenan while the other leg of the mouse was left normal as the control. As the arthritis model was established, a slight swell on the right leg of the mice was observed (Fig. 4c). The TTA-Nd-NPs based thermometer was then used to evaluate the specific temperature variations. As shown in Fig. 3c, different luminescent ratios between the two legs were clearly revealed. The ratiometric thermometry gave a clear map of temperature distributions and variations. According to the standard curve, the temperature of the arthritis area was higher than that of the normal area with an average difference of ~0.9 K. The result was in consistent with that achieved on a thermal camera (Supplementary Figure 24). Moreover, the observations also indicated that the TTA-Nd-NPs was little affected by the rigorous microenvironment in inflammation area where reactive oxygen species were relatively rich[55]. Therefore, the TTA-Nd-NPs probe is capable of thermometry in vivo for monitoring the normal and abnormal temperature variations accurately.

## Discussion

In the TTA system composed of annihilator and sensitizer dyad, an excited triplet sensitizer should diffuse to an annihilator nearby and participate in the triplet–triplet energy transfer (TTET) process to generate one triplet annihilator. Two triplet annihilators then approach each other through diffusion, and eventually annihilate to populate one singlet annihilator for the delayed TTA-UCL emission. Obviously, the diffusion is a positive factor for TTA-upconversion processes. TTA-UCL seems to become stronger at higher temperature, because the diffusion is positively correlated with temperature according to the Stokes–Einstein equation[32]. The multi-thermosensitive processes of TTA system could offer more scope for seeking a higher thermal sensitivity. Typically, TTA-upconversion was positively and highly temperature sensitive in case of blending the TTA chromophores in rubbery polymer chains[24].

However, the conventional TTA systems were incapable for thermometry in vivo due to the two problems as below. One aspect of the matter, the TTA-temperature mechanism above was non-universal when rubbery polymers was not applicable as supporting medium. For example in the system employing phase-transition (gel-to-sol) matrix, the positive temperature-dependence of TTA-UCL was no longer available[27–29]. The diffusion was approved to be a positive factor for TTA-UCL, but the negative effect of non-radiative deactivation was generally ignored. Indeed, the deactivation could lower the thermal sensitivity, or even led to a reverse response (e.g., positive at lower temperature and negative at higher temperature). Besides, without a temperature-independent standard internally, the conventional TTA system is also not qualified for thermometry in a complex biological environment[42]. Therefore, the development of TTA-based thermometry in vivo is still challenging.

The two key problems were both solved in this work. First of all, we have developed a highly thermosensitive TTA system by using the deactivation suppression strategy via molecular modification. The TTA-UCL was positively temperature-dependent over the

physiological temperature range. Subsequently, the TTA system was encapsulated together with NIR-emissive but thermal-insensitive $Nd^{3+}$ nanophosphors to construct the ratiometric TTA-Nd-NPs nanocomposite with high thermal sensitivity up to ~7.1 % $K^{-1}$. The ratiometric TTA-UCL in TTA-Nd-NPs probe could provide stable signal output in spite of the fluctuant circumstances in living system. The potential use of TTA-Nd-NPs probe as a ratiometric thermometer in vivo was demonstrated by the accurate detection of temperature distribution in tissue and temperature increment caused by inflammation in mice. We believe the findings reported herein will assist in further research on the development of ratiometric thermometer in vivo.

## Methods

**Synthesis of BDM annihilator.** Mesitaldehyde (0.734 g, 4.95 mmol) and 3-ethyl-2,4-dimethylpyrrole (1.22 g, 9.9 mmol) were dissolved in dry $CH_2Cl_2$ (150 mL) under nitrogen. One drop of trifluoroacetic acid (TFA) was added, and the resulting solution was stirred at room temperature for 6 h in the dark. Then, 2, 3-Dichloro-5, 6-dicyanoquinone (DDQ, 1.12 g, 4.95 mmol) was added to the mixture, and the solution was stirred for additional 60 min. The reaction mixture was then treated with triethylamine (3 mL) for 10 min and boron trifluoride etherate (3.4 mL) for another 2 h. The dark brown solution was washed with water (2×40 mL) and brine (40 mL), dried over anhydrous magnesium sulfate, and concentrated under reduced pressure. The crude product was purified by silica-gel flash column chromatography (elution with 10% EtOAc/petroleum ether) to yield BDM as a green crystal (yield 65%). 1H-NMR (400 MHz, $CDCl_3$) δ 6.94 (s, 2 H), 2.53 (s, 6 H), 2.34 (s, 3 H), 2.28–2.31 (q, 4 H), 2.09 (s, 6 H), 1.29 (s, 6 H), 0.99–1.01 (t, 6 H). Maldi-Tof/Tof-MS: calcd. ($[C_{26}H_{33}BF_2N_2]^+$) $m/z$ = 422.2705, found $m/z$ = 422.2708.

**Synthesis of NaYF₄: 5% Nd nanophosphors.** In a typical procedure, 6 mL oleic acid and 15 mL 1-octadecene, 0.95 mmol $YCl_3$, and 0.05 mmol $NdCl_3$ were added to a 100 mL three-necked flask. The mixture was heated to 160 °C to form a clear solution under nitrogen flow. After the solution was cooled down to room temperature, 6 mL of methanol solution containing 2.5 mmol NaOH and 4 mmol $NH_4F$ was slowly added into the flask, and stirred for about 30 min at 80 °C. Then, the solution was degassed to remove residual water and oxygen at 120 °C under vacuum for 15 min. Subsequently, the solution was heated to 300 °C and maintained for 1 h under nitrogen atmosphere. The solution was cooled naturally followed by an excessive amount of cyclohexane and ethanol were poured into. The resultant mixture was centrifugally separated, and the products were collected and washed with cyclohexane and ethanol for three times.

**Preparation of TTA-Nd-NPs.** BDM and PtTPBP were dissolved in nujol (concentrations of BDM and PtTPBP were $2 \times 10^{-3}$ mol $L^{-1}$ and $2 \times 10^{-4}$ mol $L^{-1}$, respectively) before $Nd^{3+}$ nanophosphors (15 mg $mL^{-1}$) were added in to get stock A. And BSA was dissolved in deionized water to get stock B (concentration of BSA was 1.5 mg $mL^{-1}$). Then, 1 mL stock A and 20 mg L-ascorbyl 6-palmitate were added into 100 mL stock B. The mixture was pre-emulsified at room temperature using an ultrasonic (Sonics VC750, Sonics & Materials, Inc.) for 10 min, and then was immediately emulsified using a high pressure nano homogenizer machine (FB-110Q, LiTu Mechanical equipment Engineering Co., Ltd.) at 900 bar for 20 min. The emulsion was heated at 90 °C for 1 h. Then, the emulsion was filtrated with 0.8 μm membrane to obtain sterile emulsion. Finally, the uniform nanoparticles were obtained after gradient centrifugation. The other details of materials were shown in the Supplementary Information.

**The measurement of absolute quantum efficiency.** In the experiment, the TTA-Nd-NPs material was dispersed in water at room temperature. The solution was then filled in a transparent quartz cuvette. The absolute quantum efficiency of BDM & PtTPBP in TTA-Nd-NPs was measured with Hamamatsu instrument (C13532-12 Quantaurus-QY plus). A 635-nm laser with power density of 100 mW $cm^{-2}$ was used as excitation light source. The absolute quantum efficiency of NaYF₄: 5% Nd in TTA-Nd-NPs was measured with Photon Technology International instrument (PTI QM-40). An 808-nm laser with power density of 100 mW $cm^{-2}$ was used as excitation light source. Light integrating spheres serving as accessory devices of the two instruments were used in the measurements.

**The biocompatibility of TTA-Nd-NPs probe.** The biocompatibility of TTA-Nd-NPs material was studied by using the standard methyl thiazolyl tetrazolium (MTT) assay. Hela cells ($1 \times 10^4$ per well) were pre-cultured in the 96-well plate. Subsequently, TTA-Nd-NPs at a series of concentrations (0.05, 0.1, 0.15, 0.3, 0.5, 1.0, 1.5, and 3.0 mg $mL^{-1}$) were added to the wells while DMEM as the negative control group. The cells were incubated under 5% $CO_2$ at 37 °C for 24 h. After the MTT solution was added to each well, the cells were incubated for another 4 h.

Then, an enzyme-linked immunosorbent assay reader (infinite M200, Tecan, Austria) was used to measure the cell viability.

**Bioimaging and temperature monitoring in vivo.** The animal procedures were in accordance with the guidelines of the Institutional Animal Care and Use Committee, Fudan University. UCL imaging in vivo was performed with an in vivo imaging system designed by our group, using CW 638&808-nm laser (Changchun fs-optics Co., China) as the excited source and an Andor DU897 EMCCD as the signal collector. The TTA-UCL and reference signals were acquired in channel I (485–575 nm) and channel II (980–1300 nm) of the in vivo imaging system, respectively. In consideration of the effect of excitation laser on small animal, excessive laser irradiation (e.g., continuous exposure under intense laser for long periods) should be avoided[54]. Excitation laser with low power density (100 mW $cm^{-2}$) was used to minimize the laser-induced stress and damage. The duration of laser excitation at a time was less than 10 s. The time was sufficient for acquiring images but not long enough to accumulate appreciable heat (<0.1 K). In some experiments, an IR thermal camera (FLIR E40) was used to imaging the temperature distributions on the surface of the mice.

**Data availability**. The data that support the findings of this study are available from the corresponding authors upon reasonable request.

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

## Acknowledgements

The authors thank the National Natural Science Foundation of China (21527801, 21231004, 21701109, and 21722101), the National Key R&D Program of China (Grant 2017YFA0205100 and 2016YFC130310), the National Basic Research Program of China (2015CB931800), and Shanghai Sci. Tech. Comm. (15QA1400700) for financial support.

## Author contributions

The manuscript was written by M.X, W.F., and F.L. The experiment and analysis were carried out by M.X., X.Z., Q.S., W.Y., C.C., Q.W., and X.Z. The experimental work and the manuscript were supervised by W.F. and F.L.

## Additional information

**Competing interests:** The authors declare no competing interests.

