## [Peer Review File · Nature Communications]

Reviewers' comments:

Reviewer #1 (Remarks to the Author):

IN this paper authors report on a new approach for in vivo thermometry by using a novel thermosensitive structure based on a Up-converting unit encapsulated together with a temperature independent unit. At the end the structure shows two emissions (at 550 and 1060 nm) whose intensity ratio is strongly temperature dependent. The ability of this structure for thermal sensing in in vivo conditions is demonstrated by performing a simple in vivo experiment. This is the real interesting part of the paper as authors demonstrate the possibility of measuring inflammatory processes at the in vivo level. Because of this last demonstration I think the paper should be published in NATCom but after major changes:

1.- Authors should comment on the real limitations of their sensing structure. It Works in the visible domain so that penetration depth is reduced, it needs two excitation lasers and information about its reversibility is not given. All these details and the possible solutions should be explicitly comment in the manuscript.

2.- Authors should compare the sensitivity, operating wavelengths, size, and resolution of their sensing unit to that previously reported. A possible reference paper for this comparison could be DOI: 10.1002/adom.201600508

3.- Authors should consider the heating effect of the two lasers used in the experiments. Control experiments should be given in absence of nanoparticles. Note that both excitation wavelengths are expected to produce some heating and then change the thermal measurements (doi: 10.1002/jbio.201500271).

4.- Authors should provide in the main text a comparison of their thermal image with that obtained with a reference thermal camera.

5.- Authors should provide more information about the inflammatory process. Is it time dependent? Can authors provide thermal images at different days?

6.-Note that the use of nanothermometry for the detection of inflammatory processes has been recently demonstrated in ischemic experiments. See DOI: 10.1002/adhm.201601195. This should be discussed. What are similitudes and differences?

7.- Some discussion about biodistribution should be included in the revised version.

Reviewer #2 (Remarks to the Author):

In this work, the authors presented a highly sensitive thermometer based on chromophores and Nd doped nanophosphor for biological applications. The subject under study (contactless optical nanothermometry) has received much attention from the scientific community in recent years and it is very important because it has a very broad range of applications ranging from nanomedicine, microfluidics, nanoelectronics to several others. The search for such super sensitive nanothermometers is very important because it can revolutionize all these and other areas, especially in the part of diagnosis and therapy. These information need included in the manuscript to reinforce the interest to others in the community and the wider field. Many experiments were carried out by the authors to confirm the potential of the investigated nanothermometer, including in vivo applications. The results are very interesting and deserve to be published in Nature Communications. However, in order to improve the manuscript and make it clearer, I suggest some corrections and/or clarifications.

How was measured the absolute quantum efficiency (AQE) that I believe to be absolute fluorescence quantum efficiency? This information of AQE is not in Figure S7, as pointed up by the authors.

Please, indicate each emission in the figure 2(a). The reason for the slope going from 2 to 1 was not explained by the authors and it needs to be, since a possible dependence on the excitation

power in the results is not good for nanothermometer.

For biological applications, a super important parameter is the emission wavelength and the excitation wavelength as well, because deep penetration, better emission signals, selectivity in the excitation, etc., will be better when these wavelengths are poorly or negligibly absorbed by the tissues, i.e., it is desired that only the added nanoparticles absorb this excitation light. The 635 nm excitation and 540 nm emission wavelengths are not within the biological windows. Is this not a problem? In this sense, is still this chromophore based on triplet-triplet annihilation (TTA) mechanism an interesting option? The authors need to discuss a little about it. Other question is the use of two excitation wavelengths (635 and 800 nm) and detection using two detectors (PMT and InGaAs). Is this not a problem in the experimental configuration for a practical and feasible application? This issue has also arisen because recent researches (Nano Lett. 16 (2016) 1695; Adv. Funct. Mater. 27 (2017) 1702249) are already looking for dynamic thermal imaging and these require simpler systems in configuration.

Why did the authors use Nd ions as the rare earth emitter? This ion has been shown to be not a good nanothermometer because it exhibits a low thermal sensitivity. See, for example, ACS Nano 7 (2013) 1188; Adv. Funct. Mater. 25 (2015) 615.

More recently, many other rare earth ions, along with core-shell engineering, have been presented as very efficient optical nanothermoters. See, for example, Nano Lett. 16 (2016) 1695; Adv. Funct. Mater. 27 (2017) 1702249. The authors need to comment on these recent works to some extent justify the use of the Nd ion and also to value and give credit to previous researches.

In summary, the major claim of the manuscript is the development of super sensitive nanothermometer focusing biological applications; however, it could be of great interest for many other communities. The authors also presented a new material or engineering of material for solving previous problem of biocompatibility and others.

Point-by-point response to the Reviewers' comments:

Reviewer #1

Comments: *IN this paper authors report on a new approach for in vivo thermometry by using a novel thermosensitive structure based on a Up-converting unit encapsulated together with a temperature independent unit. At the end the structure shows two emissions (at 550 and 1060 nm) whose intensity ratio is strongly temperature dependent. The ability of this structure for thermal sensing in in vivo conditions is demonstrated by performing a simple in vivo experiment. This is the real interesting part of the paper as authors demonstrate the possibility of measuring inflammatory processes at the in vivo level. Because of this last demonstration I think the paper should be published in NAtCom but after major changes:*

1.- Authors should comment on the real limitations of their sensing structure. It Works in the visible domain so that penetration depth is reduced, it needs two excitation lasers and information about its reversibility is not given. All these details and the possible solutions should be explicitly comment in the manuscript.

Reply: We greatly appreciate your insightful comment on our work. Actually, in-depth evaluation of our thermometry in the original manuscript was not sufficient. We have revised the manuscript based on your valuable suggestions. Some experimental results and discussions were added to provide a more comprehensive demonstration of our thermometry, including its advantages and real limitations.

(1) We have developed a highly sensitive *in vivo* thermometer based on TTA-upconversion system. Actually, the reduced penetration depth of visible UCL signal *in vivo* is a common issue. The issue was explicitly comment in the revised manuscript. “Frankly, the weakness of this excellent TTA-Nd-NPs thermometer was revealed as the moderate penetration depth of green UCL signal (**Figure S18**). Therefore, a thermosensitive TTA-UC system working in the NIR domain could be an attractive breakthrough for future applications in deep tissues.” Experiments had been conducted to clearly demonstrate the limitation. The tissue could have distinct effects (absorption, scattering and reflection) on emission signals of different wavelengths. Both of the two signals emitted from UCL band and NIR band became weaker under the covering of tissues at room temperature. As the thickness of tissues increased from 1 mm to 5 mm, decreasing tendency of the NIR signal was less than that of green UCL signal, which was attributed to the mitigation of tissue effect on longer-wavelength emission.

Figure S18. Luminescence intensities of TTA-Nd-NPs under the covering of tissue phantom with thickness in the range of 0–5 mm. The emission signals were acquired in the TTA-upconversion channel I (485–575 nm, UCL signal) and Nd-NIR channel II (980–1300 nm, NIR signal) of the *in vivo* imaging system, respectively. The maximum intensities of the two signals were all normalized to 1.

We were aware of the limitations and possible solutions as well. On the one hand, the bio-applications at present was focused on the subcutaneous models. On the other hand, we tried to further address the issue by an analogy method (**Figure S15**). With respect to nanothermometry, the temperature *in vivo* was deduced from a relationship between the received signals and the *in-situ* temperature. In this case, the deviations could be minimized by using a standard curve that was plotted under the same conditions as in the practical application. Therefore, the standard curve for temperature evaluation *in vivo* was measured with TTA-Nd-NPs probe in tissue phantom. The conditions of injection and imaging methods for standard curve were also the same as thermometry. Based on these strategies, we could achieve satisfying results for thermometry *in vivo*.

(2) The reversibility was also investigated and the results were added in the revised manuscript (**Figure S14**). The TTA-Nd-NPs probe was stable and showed temperature sensing ability with good reversibility. After ten times of warming-cooling cycles, just little changes happened to the emission ratios (I_{UCL}/I_{NIR}) of TTA-Nd-NPs probe.

Figure S14. The reversibility of TTA-Nd-NPs tested in continuous warming-cooling cycles. Ratiometric signals (I_{UCL}/I_{NIR}) were measured at 10 °C and 50 °C, respectively. I_{UCL} : intensities of TTA-UCL signal at 540 nm; I_{NIR} : intensities of Nd-NIR signal at 1060 nm.

Comments: 2.- Authors should compare the sensitivity, operating wavelengths, size, and resolution of their sensing unit to that previously reported. A possible reference paper for this comparison could be DOI: 10.1002/adom.201600508.

Reply: Thanks for your valuable suggestions. We have added a table with detailed parameters to compare thermometry properties of the previous studies and our work (**Table S1**). The related references were added to the Reference section in the revised manuscript (**Ref. 51**) and the revised Supporting Information (**Ref. S3–S8**).

Table S1. Experimental parameters of the *in vivo* nanothermometry reported in scientific researches.^[S3] λ_{EX} , I_{EX} , λ_{OPR} , S_r and R_{Ther} correspond to excitation wavelength, excitation laser power density, operating wavelength, relative thermal sensitivity and thermal resolution, respectively.

Material	Size (nm)	λ_{EX} (nm)	I_{EX} ($W\ cm^{-2}$)	λ_{OPR} (nm)	S_r ($10^{-2}\ K^{-1}$)	R_{Ther} (K)
GFP ^[S4]	-	473	-	480-560	1.46	-
Pbs/CdS/ZnS QDs ^[S5]	6	808	0.1-3.0	850-1650	1.25	-
LaF ₃ :Nd ³⁺ NPs ^[S6]	15	808	4	800-930	0.26	-
CsUCNP@C ^[S7]	77	730 and 980	0.3-0.8	500-580	1.10	0.5
Nd@Yb LaF ₃ NPs ^[S8]	24	808	0.7	900-1360	0.4	1
TTA-Nd-NPs ^[this work]	165	635 and 808	0.1	540 and 1060	7.1	0.1

Comments: 3.- Authors should consider the heating effect of the two lasers used in the experiments. Control experiments should be given in absence of nanoparticles. Note that both excitation wavelengths are expected to produce some heating and then change the thermal measurements (doi: 10.1002/jbio.201500271).

Reply: Thanks for your valuable suggestions. A control experiment was conducted in absence of TTA-Nd-NPs nanoparticles. In the experiment, a Kunming mouse was irradiated with 635-nm and 808-nm lasers, respectively. The temperatures at different time points were recorded with an IR thermal camera and the results were added in the revised manuscript. Surface temperature of the mouse showed no incensement at the first 10 s (< 0.1 K, **Figure S21**), under irradiation of whether 635-nm laser (right, 100 mW cm^{-2}) or 808-nm laser (left, 100 mW cm^{-2}). As for experimental conditions in the manuscript, the laser exposure below this level could keep influence on thermometry at a minimum. In addition, a temperature increment of ~ 0.5 K was also observed under continuous irradiation of 808-nm laser for 60 s. Therefore, in the revised manuscript we have added some details in the Method section to clarify the *in vivo* experiments. “In consideration of the effect of excitation laser on small animal, excessive laser irradiation (e.g. continuous exposure under intense laser for long periods) should be avoided.⁵⁵ Excitation laser with low power density (100 mW cm^{-2}) was used to minimize the laser-induced stress and damage. The duration of laser excitation at a time was less than 10 s. The time was sufficient for acquiring images but not long enough to accumulate appreciable heat (< 0.1 K).” The related reference was also added to the Reference section in the revised manuscript (**Ref. 54**).

Figure S21. The possible influence of excitation laser on thermometry. A Kunming mouse was irradiated with 635-nm and 808-nm lasers, respectively. The temperatures at different time points were recorded with an IR thermal camera. Surface temperature of the mouse showed no incensement at the first 10 s (< 0.1 K), under irradiation of whether 635-nm laser (right, 100 mW cm^{-2}) or 808-nm laser (left, 100 mW cm^{-2}). As for experimental conditions in the manuscript, the laser exposure below this level could keep influence on thermometry at a minimum.

Comments: 4.- Authors should provide in the main text a comparison of their thermal image with that obtained with a reference thermal camera.

Reply: Thanks for your kind suggestions. Apart from the luminescent nonthermometry, the temperature was also measured with an IR thermal camera. As revealed in **Figure S22**, the temperature variation between two legs was 0.9 K, which was in consistent with the result obtained with TTA-Nd-NPs thermometry (**Figure 3** in the manuscript). The related discussions were added in the main test of the revised manuscript.

Figure S22. The inflammation-induced temperature variation. The bright-field image (a) showed arthritis in the swollen right leg of a Kunming mouse stimulated with carrageenan while the left leg as control was normal. The temperature distributions were measured with a thermal camera (b).

Comments: 5.- Authors should provide more information about the inflammatory process. Is it time dependent? Can authors provide thermal images at different days?

Reply: Thanks for your valuable suggestions. The inflammatory process was investigated in a mild inflammation model. The photos and thermal images at different time points were provided (**Figure S19**). After the inflammation model was established in a Kunming mouse (5 h post-injection of carrageenan), the photos showed that the left leg was swollen while the right leg as control was normal. As revealed in the thermal images, the

temperature in the left leg was higher than that of the right leg ($\Delta T=0.7$ K). The temperature deviation between two legs (ΔT) came up to ~ 1 K at 12 h. Then, the detumescence of left leg was observed along with the decreasing ΔT values. About three days later, the mild inflammation was almost healed up because the morphology and temperature of the left leg became normal. Therefore, the results revealed that the inflammation was time dependent.

Figure S19. The photos and thermal images of inflammation model at different time points. The left leg of a Kuming mouse was inflamed while the right leg as control was normal. The carrageenan was used to induce mild arthritis in the left leg.

Comments: 6.-Note that the use of nanothermometry for the detection of inflammatory processes has been recently demonstrated in ischemic experiments. See DOI: 10.1002/adhm.201601195. This should be discussed. What are similarities and differences?

Reply: Thanks for your valuable suggestions. The use of nanothermometry for the detection of ischemia and inflammation was significant for clinical diagnosis and therapy. Therefore, this information was integrated into the Introduction section and the related reference was added to the Reference section in the revised manuscript (**Ref. 4**). In addition, we discussed the work of ischemia experiments in the revised manuscript to value and give credit to this meaningful research. “The inflammatory phenomena has been demonstrated in distinct processes, such as the ischemia symptoms, physical trauma, microbial infection, and drug induction.^{4,52,53} Previously, the nanothermometry results revealed that the inflammatory process in ischemia experiments could lead to distinctive thermal dynamics in mice.⁴ With these in mind, we intended to establish a chemically-

induced arthritis with carrageenan and to monitor the inflammation-based temperature fluctuations directly (**Figure S19**). Notably, we demonstrated that the inflammation in arthritis model was time-dependent and was accompanied with temperature variations (**Figure S20**).” Attributed to immune system in the mouse, the mild inflammation induced by carrageenan was almost healed up in three days and the temperature of left leg became normal.

Figure S20. The temperature deviations (ΔT) at different time points. In the inflammation model, the left leg of a Kuming mouse was inflamed while the right leg as control was normal. ΔT is the temperature deviations between two legs, namely $\Delta T = T_{\text{left}} - T_{\text{right}}$.

Comments: 7.- *Some discussion about biodistribution should be included in the revised version.*

Reply: Thanks for your kind suggestions. We have added some discussion about biodistribution of the TTA-Nd-NPs (**Figure S12**). After intravenous injection of TTA-Nd-NPs, the nanoparticles were quickly accumulated in the liver as revealed from the strong TTA-UCL emission. Bioimages showed high accumulation of TTA-Nd-NPs in the liver and spleen, while no appreciable TTA-UCL signal from the kidney or lung was detected. The accumulation of TTA-Nd-NPs in liver and spleen was attributed to the abundant macrophages in these organs that reinforced the uptake and metabolism of foreign particles. The results and discussions were added in the revised manuscript.

Figure S12. TTA-UCL intensities of mice organs at various time points post-injection. The mice were intravenously injected with TTA-Nd-NPs. The emission signals were acquired in the TTA-upconversion channel I (485–575 nm) of the bioimaging system.

Reviewer #2

Comments: *In this work, the authors presented a highly sensitive thermometer based on chromophores and Nd doped nanophosphor for biological applications. The subject under study (contactless optical nanothermometry) has received much attention from the scientific community in recent years and it is very important because it has a very broad range of applications ranging from nanomedicine, microfluidics, nanoelectronics to several others. The search for such super sensitive nanothermometers is very important because it can revolutionize all these and other areas, especially in the part of diagnosis and therapy. These information need included in the manuscript to reinforce the interest to others in the community and the wider field. Many experiments were carried out by the authors to confirm the potential of the investigated nanothermometer, including in vivo applications. The results are very interesting and deserve to be published in Nature Communications. However, in order to improve the manuscript and make it clearer, I suggest some corrections and/or clarifications.*

Reply: We are grateful your valuable comments and suggestions. We agree with you that the significance of contactless optical nanothermometry should be emphasized to reinforce the interest to others in the community and the wider field. According to your suggestion, we have added these relevant information in the background introduction of the revised manuscript. “The luminescent nanothermometry has received much attention in recent years because it has a broad range of applications involving nanomedicine,

microfluidics, nanoelectronics and integrated photonic devices.¹⁻³ The development of such highly sensitive nanothermometer is very important in view of its great potential to revolutionize relevant areas, especially in the part of diagnosis and therapy.⁴” Thanks for your kind help to improve our manuscript.

Comments: *How was measured the absolute quantum efficiency (AQE) that I believe to be absolute fluorescence quantum efficiency? This information of AQE is not in Figure S7, as pointed up by the authors.*

Reply: Thanks for your valuable comment. The measurement details of absolute quantum efficiency was added to Methods section in the revised manuscript. The sentence in the revised manuscript is as below. “The measurement of absolute quantum efficiency: In the experiment, the TTA-Nd-NPs material was dispersed in water at room temperature. The solution was then filled in a transparent quartz cuvette. The absolute quantum efficiency of BDM & PtTPBP in TTA-Nd-NPs was measured with Hamamatsu instrument (C13532-12 Quantaaurus-QY plus). A 635-nm laser with power density of 100 mW cm⁻² was used as excitation light source. The absolute quantum efficiency of NaYF₄: 5% Nd in TTA-Nd-NPs was measured with Photon Technology International instrument (PTI QM-40). An 808-nm laser with power density of 100 mW cm⁻² was used as excitation light source. Light integrating spheres serving as accessory devices of the two instruments were used in the measurements.” Thank you for pointing out our mistake.

Comments: *Please, indicate each emission in the figure 2(a). The reason for the slope going from 2 to 1 was not explained by the authors and it needs to be, since a possible dependence on the excitation power in the results is not good for nanothermometer.*

Reply: Thanks for your valuable suggestions. We have revised the manuscript to make these clearer.

(1) According to your suggestion, the emission information was indicated in the caption of Figure 2a. The descriptive sentence in the revised manuscript is “TTA-UCL signal peaked at 540 nm and Reference signal peaked at 1060 nm were plotted in the emission profiles”.

(2) According to your suggestion, the reason for the slope going from 2 to 1 was also explained. The explanatory sentence in the revised manuscript is “Under low-power excitation below the threshold, triplets of the annihilators decay spontaneously to result in a quadratic dependence, which is a common phenomenon in the bimolecular TTA-upconversion system.⁴³”.

(3) We agree with you that a possible dependence on the excitation power in the results is not good for nanothermometer. Therefore, in this work we have managed to avoid the possible dependence. In our experiments, power densities of coupled 635 & 808 nm excitation light were set beyond the power threshold ($>65 \text{ mW cm}^{-2}$) to ensure both of the luminescence processes in a linear regime. We have further elaborated the issue in the revise manuscript. “Herein, laser power density in the range of 100–200 mW cm^{-2} was an optimal choice for excitation, which could enable power-independent signal output in the low-power region (Figure 2a).”

Comments: *For biological applications, a super important parameter is the emission wavelength and the excitation wavelength as well, because deep penetration, better emission signals, selectivity in the excitation, etc., will be better when these wavelengths are poorly or negligibly absorbed by the tissues, i.e., it is desired that only the added nanoparticles absorb this excitation light. The 635 nm excitation and 540 nm emission wavelengths are not within the biological windows. Is this not a problem? In this sense, is still this chromophore based on triplet-triplet annihilation (TTA) mechanism an interesting option? The authors need to discuss a little about it. Other question is the use of two excitation wavelengths (635 and 800 nm) and detection using two detectors (PMT and InGaAs). Is this not a problem in the experimental configuration for a practical and feasible application? This issue has also arisen because recent researches (Nano Lett. 16 (2016) 1695; Adv. Funct. Mater. 27 (2017) 1702249) are already looking for dynamic thermal imaging and these require simpler systems in configuration.*

Reply: Thanks for your valuable comments. The multi-thermosensitive processes of TTA system could offer more scope for seeking a higher thermal sensitivity. The theory was improved and elaborated in the Discussion section of the manuscript. In the present work, we have developed a highly sensitive *in vivo* thermometer based on TTA-upconversion system (**Table S1**). Actually, the reduced penetration depth of visible UCL signal *in vivo* was a common issue and the system in configuration was not simple enough at present. Indeed, we were aware of the limitations and possible solutions as well. The issues were explicitly comment in the revised manuscript. “Frankly, the weakness of this excellent TTA-Nd-NPs thermometer was revealed as the moderate penetration depth of green UCL signal (Figure S18). Therefore, a thermosensitive TTA-UC system working in the NIR domain could be an attractive breakthrough for future applications in deep tissues. The NIR emissive sensitizer severing as reference could also simplify the systems in configuration.^{46,47}” The related references were also added to the Reference section in the revised manuscript (**Ref. 46** and **Ref. 47**). In this work, we have managed to establish the installations (**Figure 2b** and **Figure 3a**) and analogy methods (**Figure S15**) for ratiometric thermometry. Based on these efforts and strategies, we have achieved satisfying results for thermometry *in vivo*.

Table S1. Experimental parameters of the *in vivo* nanothermometry reported in scientific researches.^[S3] λ_{Ex} , I_{Ex} , λ_{Opr} , S_r and R_{Ther} correspond to excitation wavelength, excitation laser power density, operating wavelength, relative thermal sensitivity and thermal resolution, respectively.

Material	Size (nm)	λ_{Ex} (nm)	I_{Ex} (W cm^{-2})	λ_{Opr} (nm)	S_r (10^{-2} K^{-1})	R_{Ther} (K)
GFP ^[S4]	-	473	-	480-560	1.46	-
Pbs/CdS/ZnS QDs ^[S5]	6	808	0.1-3.0	850-1650	1.25	-
LaF ₃ :Nd ³⁺ NPs ^[S6]	15	808	4	800-930	0.26	-
CsUCNP@C ^[S7]	77	730 and 980	0.3-0.8	500-580	1.10	0.5
Nd@Yb LaF ₃ NPs ^[S8]	24	808	0.7	900-1360	0.4	1
TTA-Nd-NPs ^[this work]	165	635 and 808	0.1	540 and 1060	7.1	0.1

Comments: *Why did the authors use Nd ions as the rare earth emitter? This ion has been shown to be not a good nanothermometer because it exhibits a low thermal sensitivity. See, for example, ACS Nano 7 (2013) 1188; Adv. Funct. Mater. 25 (2015) 615.*

Reply: Thanks for your valuable comments. We agree with you that Nd-based nanothermometer has low thermal sensitivity. Therefore it is used to provide temperature-independent inner standard here to generate ratiometric signal. In the revised manuscript, we have further elaborated the issue and our design concept in terms of the issue as well. “In contrast, the emission from Nd³⁺ nanocrystals in TTA-Nd-NPs slowly declined with a slope less than 0.03% K⁻¹ (**Figure 2c**). Indeed, the Nd ion was not good for thermal response due to its low thermal sensitivity.⁴⁴⁻⁴⁵ Attributed to the highly thermosensitive nature of our TTA system that was served as temperature-responsive unit, herein the thermal insensitive Nd³⁺ nanocrystal could be designed to just serve as an internal standard. The absorption/emission of Nd³⁺ nanocrystals showed no overlap with that of the TTA system, which enabled the unaffected performance of TTA system.” The related references were also added to the Reference section in the revised manuscript (**Ref. 44** and **Ref.45**).

Comments: *More recently, many other rare earth ions, along with core-shell engineering, have been presented as very efficient optical nanothermoters. See, for example, Nano Lett. 16 (2016) 1695; Adv. Funct. Mater. 27 (2017) 1702249. The authors need to comment on these recent works to some extent justify the use of the Nd ion and also to value and give credit to previous researches.*

Reply: Thanks for your kind suggestions. According to your suggestions, we have added these information in the revised manuscript. “Actually, calibration unit for the TTA

system can also be upgraded in the future, for example employing other rare earth ions based core-shell nanoparticles that showed markedly thermal sensitivity, which is possibly beneficial for achieving a better performance in ratiometric thermometry.⁴⁶⁻⁴⁷ The related references were also added to the Reference section in the revised manuscript (**Ref. 46** and **Ref.47**).

Comments: *In summary, the major claim of the manuscript is the development of super sensitive nanothermometer focusing biological applications; however, it could be of great interest for many other communities. The authors also presented a new material or engineering of material for solving previous problem of biocompatibility and others.*

Reply: Thanks for your insightful summary of our work. We really appreciate the valuable suggestions from your professional perspective which help us improved the manuscript. We believe the revised manuscript will make more sense to the related researches.

Again, we are grateful to all the constructive comments made by the reviewers and the editorial office.

Sincerely,

Wei Feng

REVIEWERS' COMMENTS:

Reviewer #1 (Remarks to the Author):

Paper is ready for publication as authors have satisfied all my previous requirements

Reviewer #2 (Remarks to the Author):

The work was greatly improved according to the referees' comments.

I just expected a little more emphasis and appreciation of the work in the introduction, because I really see that the work is broader, covers more areas. The authors copied only what I put as a comment and it was not just for that, it was just a suggestion. However, that is fine.

I recommend accepting the work for publication!

Point-by-point response to the Reviewers' comments:

Reviewer #1

Comments: *Paper is ready for publication as authors have satisfied all my previous requirements.*

Reply: Thank you very much for your kind effort to improve our manuscript.

Reviewer #2

Comments: *The work was greatly improved according to the referees' comments.*

I just expected a little more emphasis and appreciation of the work in the introduction, because I really see that the work is broader, covers more areas. The authors copied only what I put as a comment and it was not just for that, it was just a suggestion. However, that is fine.

I recommend accepting the work for publication!

Reply: We are grateful for your valuable comments and suggestions. To further emphasize significance of the work, we have added one more sentence in the introduction of the revised manuscript. "The work makes great sense for a broad research areas of upconversion, thermometry, nanomedicine and life science."

Again, we are grateful to all the constructive comments made by the reviewers and the editorial office.

Sincerely,

Wei Feng